# GPy-ABCD: A Configurable Automatic Bayesian Covariance Discovery Implementation

**T. Fletcher**                                   T.Fletcher-6@sms.ed.ac.uk
*The University of Edinburgh*

**A. Bundy**                                              A.Bundy@ed.ac.uk
*The University of Edinburgh*

**K. Nuamah**                                           K.Nuamah@ed.ac.uk
*The University of Edinburgh*

## Abstract

*Gaussian Processes (GPs) are a very flexible class of nonparametric models frequently used in supervised learning tasks because of their ability to fit data with very few assumptions, namely just the type of correlation (kernel) the data is expected to display. Automatic Bayesian Covariance Discovery (ABCD) is an iterative GP regression framework aimed at removing the requirement for even this initial correlation form assumption. An original ABCD implementation exists and is a complex stand-alone system designed to produce long-form text analyses of provided data. This paper presents a lighter, more functional and configurable implementation of the ABCD idea, outputting only fit models and short descriptions: the Python package GPy-ABCD, which was developed as part of an adaptive modelling component for the FRANK query-answering system. It uses a revised model-space search algorithm and removes a search bias which was required in order to retain model explainability in the original system.*

## 1. Introduction

Automatic Bayesian Covariance Discovery (ABCD) (Lloyd, James Robert et al., 2014) is an unsupervised learning system which iteratively runs Gaussian Process (GP) regressions in order to select the best fitting model within some functional-form search-space limits. Its noteworthy utility lies not just in the generally close fit one can expect from it, but in the high level of interpretability of its outputs, which can capture even functional forms which vary over the given domain. An example for clarity: a pattern which an ABCD system is able to identify and describe in text would be (paraphrasing to shorten it) "the data starts as a linear function but then begins a periodic quadratic growth". The original ABCD implementation (now the main constituent of the Automatic Statistician (Steinruecken, Christian et al., 2019)), is a large project with parts written in MATLAB, Python, Mathematica, Fortran and more, and the outputs it produces are very detailed multi-page, text-based analyses of the given data, describing contributions from each component of the identified modular functional form. This paper describes a more functional and highly configurable ABCD system implementation which focusses on improving the model-space search and not on the output analysis, instead returning just the models and 1-paragraph descriptions. GPy-ABCD is an open-source Python package (Fletcher, 2020) built on the GPy library developed by the Sheffield machine learning group (2012); it is intended as an adaptive modelling tool to be used within a broader analysis workflow focussing on

interpretability. In particular, the workflow it was developed for is that of a statistics expert system within the Functional Reasoning for Acquiring Novel Knowledge (FRANK) Query-Answering (QA) system (Nuamah et al., 2016; Bundy et al., 2018).

## 2. Background

Grosse, Roger et al. (2012) noted (as others had before) that many common probabilistic models can be represented as compositions of simpler ones, and they used matrix decomposition models to construct a context-free grammar generating a compositional model space. Duvenaud, David et al. (2013) built on this work and created the proof-of-concept which later became the ABCD system (Lloyd, James Robert et al., 2014). Extensions to this research include joint data modelling (separating common features from individual ones) (Tong and Choi, Jaesik; Hwang, Yunseong and Choi, Jaesik, 2015), model criticism (Lloyd, James Robert and Ghahramani, Zoubin, 2015) and Bayesian corrections to final-model variance (Janz, David et al., 2016). The original ABCD became part of the Automatic Statistician (Riaz Moola; Kim, Hyunjik and Teh, Yee Whye, 2017; Steinruecken, Christian et al., 2019), and its use in classification tasks was also explored (Nikola Mrkšić, 2014). GPy-ABCD is based on the Automatic Statistician's ABCD and was designed as a simple and developer-friendly modelling tool since its functionality was required within the FRANK QA system (Nuamah et al., 2015, 2016; Nuamah and Bundy, 2019, 2018; Bundy et al., 2018). Briefly, FRANK fits in the so-called "Third wave of AI", employing both symbolic and statistical reasoning to answer data-focussed queries, e.g. "Will the African country with highest GDP in 2040 have a higher population than the equivalent South-American one?", which requires language parsing, online data sourcing, symbolic reasoning and statistical modelling to answer. In particular, GPy-ABCD was created to be used by FRANK's statistics expert system Statistical Methodology Advisor at Reasoning Time (SMART), whose purpose is to perform appropriate statistical procedures based on specific query and data properties. In this context GPy-ABCD would be selected in place of other statistical methods for queries involving univariate functional shape description (e.g. "How does rainfall in the UK behave over time?" or "Is population growth in Asia periodic/linear?"). GPy-ABCD is concerned with re-implementing only the simplest-output ABCD functionality, as that is what is required by FRANK. Comparing the respective broader systems, i.e. the Automatic Statistician and FRANK, the former is concerned with producing in-depth analyses of directly provided data of specific kinds, while the latter is a general-purpose QA system automating data procurement and method choice.

## 3. The ABCD Idea

Parametric regression methods are defined by strict assumptions on the nature of the data being analysed, i.e. the form of the predicting function the parameters of which they tune (e.g. a polynomial of a specific order). Nonparametric models, on the other hand, place much lower restrictions on the predicting function's form, using the data itself to adjust it, making them ideal candidates for learning tasks. Standard examples of nonparametric modelling methods are Support Vector Machines, GPs and different variations of Splines, all of which still require some initial assumptions to restrict their functional forms. In the case

of GPs (briefly covered in Appendix A) the assumption is the choice of covariance function the data is expected to exhibit, and although the generality of some common kernels allows extreme fitting flexibility, what is lost in exchange is a level of data interpretability, making kernel choice still subject to the modeller's judgement prior to fitting. ABCD is an iterative GP regression framework which explores a space of modularly-constructed kernels in order to identify the ones which best balance closeness of fit and expression complexity, thus reducing the required modelling assumptions even further. As laid out in (Lloyd, James Robert et al., 2014), the core components of this framework are the following:

1. An open-ended and expressive language of models
2. An efficient generation and search procedure to explore the model space
3. A model evaluation and comparison method balancing complexity and closeness of fit
4. A procedure to automatically generate descriptions of the best candidate(s)

## 4. Implementation

The following sub-sections cover GPy-ABCD's components as outlined above; they prioritise abstract description over specific details since each section is mirrored by one in Appendix B, where differences from the original ABCD are also explained.

### 4.1 The Model Language

The model language is defined by a context-free grammar constituted by a specific selection of base GP kernels which can be combined by addition and multiplication to produce more complex modular kernels. The criterion with which the base kernels were selected was to cover data features which are both common and simple enough to be easily interpretable, resulting in the following set (Duvenaud, David et al., 2013) (details in Section B.1):

- White Noise (WN) kernel to model uncorrelated noise
- Constant (C) kernel to model constant functions (useful for simple mean shifts)
- Linear (LIN) kernel to model linear functions and, when repeatedly multiplied by itself, higher order polynomials
- Squared Exponential (SE) kernel to model generic smooth functions
- (Generalised) Periodic (PER) kernel to model generic periodic functions

The SE kernel is present but disabled by default in GPy-ABCD since its versatility and small number of parameters make it too competitive against more complex but more descriptive expressions. Figure 1 shows examples of curves from a simple multiplication of kernel expressions. The grammar also contains two additional kernel operators built on lower-level sigmoidal kernels (see Appendix B.1):

- Change-Point (CP) operator: $CP[k_1, k_2] = S \times k_1 + Sr \times k_2$
- Change-Window (CW) operator: $CW[k_1, k_2] = SI \times k_1 + SIr \times k_2$

These allow the inclusion in the language of models which permanently or temporarily change covariance form, e.g. from linear to periodic (examples in Figure 2 ).

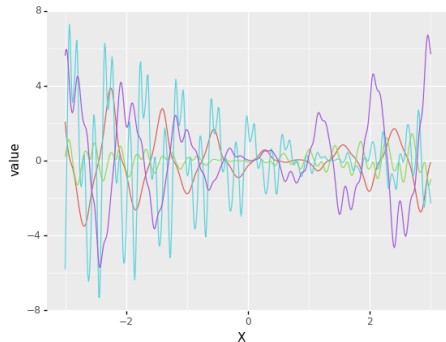

Figure 1: $LIN \times PER$: periodic functions with linearly varying amplitudes

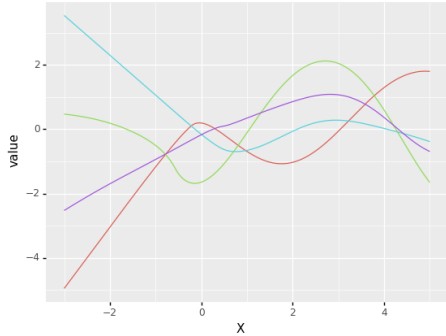

Figure 2: $CP(LIN, PER)$: linear functions becoming periodic

## 4.2 Kernel Expressions & Simplification

The core of GPy-ABCD is the `KernelExpression` class, which is the symbolic representation of kernels: a non-trivial kernel is a tree of `KernelExpression` nodes which represent kernel operations; their base-kernel arguments are contained directly in the node, while more complex arguments are in children nodes. For example $LIN \times (PER + C)$ has two nodes: a `ProductKE` root containing the factor $LIN$ and a child `SumKE` containing $PER$ and $C$. The kernel expression classes have methods providing the following general functionality:

- They self-simplify when modified
- They can produce GPy kernel objects
- They can extract the fit model parameters from a matching GPy object
- They can rearrange to a (canonical) sum-of-products form
- They can generate text descriptions of their sum-of-products form

Simplification of Kernel Expressions (KEs) is just mathematical rearrangement into more succinct forms by some basic rules (see Section B.2). Here are two simplification examples for clarity: $LIN \times PER \times WN \rightarrow LIN \times WN$ and $(SE \times SE + LIN) \times C \rightarrow SE + LIN$. These simplifications are self-triggering and take place both before model fitting and when later reducing to canonical sum-of-products form.

## 4.3 The Model-Space Search

GPy-ABCD's model-space search algorithm is a key difference from the original ABCD; it is essentially a configurable beam search (a limited-bandwidth best-first-search) using a context-free grammar as the successor states' generator. The overall algorithm is visualised in Figure 3 and explained below, with the main configurable inputs underlined; a more detailed description making reference to all inputs of the main search function is provided in Section B.3. After fitting an initial (heuristic) list of simple KEs, a predetermined number (M) of search rounds is performed; each round consists of the expansion (by grammar production rules) of the N best-fitting KEs so far, which are then simplified and filtered for duplicates before fitting (the simplified KEs do not need to be in sum-of-products

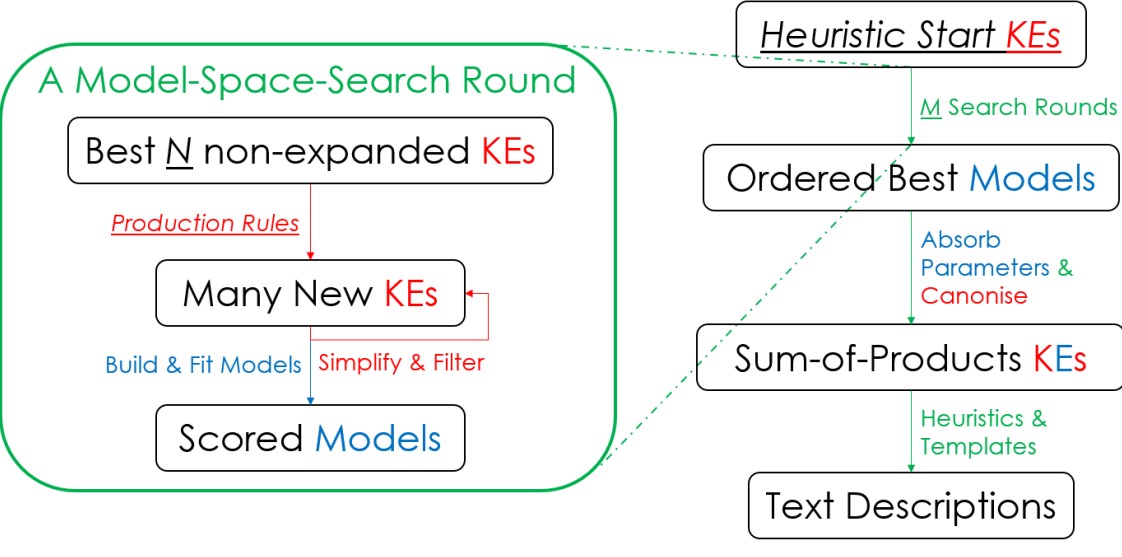

Figure 3: The model-space search algorithm

form in order to retain final model interpretability, which was a limitation of the original ABCD; see Section B.2). Each model is scored by a utility function balancing closeness of fit with expression complexity (see Section B.3.1 for a discussion on statistical issues and possibilities). The main algorithm output is then the ordered list of best fitting models.

### 4.4 Model Description

By converting a fit kernel to its canonical sum-of-product form, each product's factors are ordered by a fixed pattern: PER, WN, SE, C, LIN and then any sigmoidal kernel. This is done in order to assign fixed roles to specific kernels if present in particular combinations; the first available kernel takes the role of principal functional form, and all the others of its modifiers. The sentence production is then performed by simple templates which describe each kernel differently according to their role and which fill-in parameter values. Tables detailing these combinations are available in the ABCD literature (Lloyd, James Robert et al., 2014), but an example output for $LIN \times (PER + C)$ would be:

> "The fit kernel consists of 2 components:
> - A linear function with offset -0.09; this component has overall variance 1.04
> - A periodic function with period 6.24 and lengthscale 1774.03, with linearly varying amplitude with offset -0.09; this component has overall variance 0.54"

where the "2 components" are from distributing the product. This system works well because the base kernels were chosen from the beginning with the purpose of interpretability.

### 5. Evaluation

The base hypothesis under evaluation for GPy-ABCD is that it behaves as expected, i.e.:

- That it recognises the correct patterns in synthetic data as an ABCD system should (i.e. that its kernels do indeed work individually and together)
- That it fits and describes complex data similarly enough (taking into account the algorithm difference) to the original ABCD system implementation

Then the main hypothesis is that it can fulfil the role it was originally designed for in the FRANK QA system, i.e. being able to provide the data required to introduce the new types of queries mentioned in Section 2. Appendix C expands on the below summary.

The $1^{st}$ sub-point was evaluated by trying to recover known kernels when fitting noisy data produced from a set of formulae of varying complexities (e.g. the "obvious" match for $y = 2x \cos(\frac{x-5}{2})^2$ is $LIN \times (PER + C)$). The system identified the intended kernel in all cases (though not necessarily as the top result) except for the class of CW kernels with non-stationary first-arguments (e.g. $CW(LIN, \cdots)$); see Section C.1 for a discussion on the reasons. The $2^{nd}$ sub-point was evaluated by trying to replicate the core outputs of the original ABCD system on datasets in their literature. Although no noticeable differences in closeness of fit were present, the KEs identified by the two systems were broadly similar but not matching, with GPy-ABCD's expressions being generally simpler than the original's; this is reasonable since the comparison setup tried to match the number of rounds of the respective algorithms and not their search depth (see Section C.2), but the effects of the differences in implementation and underlying frameworks and fitting libraries are an unquantified factor. With regards to the main point of evaluation, GPy-ABCD is indeed able to provide FRANK with the means to implement the target functionality, and the computation-time constraints of the QA context stimulated the development in both projects of useful infrastructure to control the model-space search.

## 6. Conclusion

GPy-ABCD is a working implementation of an ABCD modelling system, replicating and improving the core components of the original one (i.e. not the production of detailed analyses); comparing the two, GPy-ABCD stands out for the improved model-search algorithm and its extensive customisability. Though there is room for further numerical method improvements and investigation of search-path behaviour differences from the original implementation, the system behaves as expected on both synthetic and real data, and can help users identify and describe patterns in the exploratory data analysis phase of research. At the time of writing the library has been downloaded over 10000 times. Partly due to having been developed to serve as one of many components in a broader statistics expert system, GPy-ABCD constitutes a solid base on which to build further functionality and research: an identified issue of statistical appropriateness of the choice of utility function (Section B.3.1) is the primary theory-based next step, while more practical expansion directions include extending the system to handle multidimensional data and adding more fitting methods (two things which GPy is already equipped for), as well as further increasing user customisation to encompass providing arbitrary base kernels, since specific scenarios and optimisations may require broader or narrower selections to work well.

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

## Appendix A. Gaussian Process Regression

**Definition 1** *(Rasmussen and Williams, 2006) A **Gaussian Process (GP)** is a collection of random variables, any finite number of which have a joint Gaussian distribution.*

A GP $f$ is fully defined by its mean function $m$ and covariance ("kernel") function $k$ $(f(\boldsymbol{x}) \sim \mathcal{GP}(m(\boldsymbol{x}), k(\boldsymbol{x}, \boldsymbol{x}')))$, but given the Bayesian nature of the fitting process, specifically the fact that a mean-0 prior does not limit the posterior mean to 0, it is common to assume the prior mean to be 0 and therefore let GPs be completely specified by their covariance function, which also simplifies notation. From the simple description above it should be clear that a GP regression does not fit an analytic expression for the given data but instead one for the process which could have produced it (accounting for noise): the tuned parameters are for the (covariance of the) distribution from which the input could have been sampled. Consequently predictions made from a GP model are not simple points or vectors with errors, but individual univariate or multivariate Gaussian distributions. $k(\boldsymbol{x}, \boldsymbol{x}')$ is usually also referred to as the GP's "kernel", and in typical GP regression scenarios the key feature a user is looking for in their kernel is generality, leading many applications to use the very flexible SE kernel (often also referred to as Radial Basis Function kernel): $k_{SE}(\boldsymbol{x}, \boldsymbol{x}') = \sigma^2 \exp\left(-\frac{(\boldsymbol{x}-\boldsymbol{x}')^2}{2l^2}\right)$, where $\sigma^2$ is the variance and $l$ is the lengthscale. Variance and lengthscale are parameters shared by many kernels, and intuitively, taking the kernel as a description of similarity between data observations, the variance regulates the average distance from the whole process mean, while the lengthscale regulates the average length of the fit curve's undulations (also serving as a gauge of reliable extrapolation distance). The above being granted, the choice of kernel is instead at times the crucial step in one's analysis in order to match the given data features or known context since it determines the generalisation properties of the resulting GP model; this is the task which an ABCD system (Section 3) is meant to perform in place of human analysts. The capabilities of the GPy Python library amply cover the features which an ABCD back-end requires.

## Appendix B. Details & Comparison

The following sub-sections mirror the structure of Section 4, further exploring implementation details and comparing them with the original ABCD.

### B.1 The Model Language

The base kernel expressions are reported below, where $\sigma^2$ denotes variance:

- $WN(\boldsymbol{x}, \boldsymbol{x}') = \sigma^2 \delta_{\boldsymbol{x},\boldsymbol{x}'}$, where $\delta_{\boldsymbol{x},\boldsymbol{x}'}$ is the Kronecker delta function
- $C(\boldsymbol{x}, \boldsymbol{x}') = \sigma^2$
- $LIN(\boldsymbol{x}, \boldsymbol{x}') = \sigma^2(\boldsymbol{x}-\boldsymbol{c})(\boldsymbol{x}'-\boldsymbol{c})$, where $\boldsymbol{c}$ is the horizontal offset (or a polynomial root when in repeated multiplications)
- $SE(\boldsymbol{x}, \boldsymbol{x}') = \sigma^2 \exp\left(-\frac{(\boldsymbol{x}-\boldsymbol{x}')^2}{2l^2}\right)$

- $PER(\boldsymbol{x}, \boldsymbol{x'}) = \sigma^2 \dfrac{\exp\left(\dfrac{\cos\left(\frac{2\pi(\boldsymbol{x}-\boldsymbol{x'})}{p}\right)}{l^2}\right) - I_0\left(\frac{1}{l^2}\right)}{\exp\left(\frac{1}{l^2}\right) - I_0\left(\frac{1}{l^2}\right)}$, where $p$ is the period and $I_0$ is the modified Bessel function of the first kind of order zero

The sigmoidal kernels used by the change operators are constructed as follows:

- choosing a sigmoidal function $sig$ (the used one is $\frac{x}{1+|x|}$)
- one can define a step-function (and its reverse) by scaling $sig$ to the range $[0, 1]$: $\sigma(\boldsymbol{x}) = \frac{1}{2}\left(1 + sig(\frac{\boldsymbol{x}-\boldsymbol{l}}{s})\right)$ and $\overline{\sigma}(\boldsymbol{x}) = 1 - \sigma(\boldsymbol{x}) = \sigma(\boldsymbol{x}; s \to -s)$, where $l$ and $s$ are location and slope parameters
- one can then also define sigmoidal versions of indicator functions: $hat(\boldsymbol{x}) = \frac{\sigma(\boldsymbol{x}) + \overline{\sigma}(\boldsymbol{x}+\boldsymbol{w}) - 1}{h(\boldsymbol{w})}$ and $well(\boldsymbol{x}) = \frac{2 - \sigma(\boldsymbol{x}) - \overline{\sigma}(\boldsymbol{x}+\boldsymbol{w})}{h(\boldsymbol{w})}$, where $\boldsymbol{w}$ is the (strictly positive) window width and $h(\boldsymbol{w})$ is the maximum numerator value (i.e. the hat height or well depth), scaling the expressions to 1

The sigmoidal kernels are then straightforward to construct:

- Sigmoidal (S) and Reverse Sigmoidal (Sr) kernels:
$S(\boldsymbol{x}, \boldsymbol{x'}) = \sigma(\boldsymbol{x})\sigma(\boldsymbol{x'})$ and $Sr(\boldsymbol{x}, \boldsymbol{x'}) = \overline{\sigma}(\boldsymbol{x})\overline{\sigma}(\boldsymbol{x'})$
- Sigmoidal Indicator (SI) and Reverse Sigmoidal Indicator (SIr) kernels:
$SI(\boldsymbol{x}, \boldsymbol{x'}) = hat(\boldsymbol{x})hat(\boldsymbol{x'})$ and $SIr(\boldsymbol{x}, \boldsymbol{x'}) = well(\boldsymbol{x})well(\boldsymbol{x'})$

Only WN, C and SE are already present in GPy, while the rest had to be newly implemented; more specifically, GPy does provide versions of LIN and PER, and both were tested but eventually discarded in favour of re-implementations of the original ABCD's. GPy's LIN kernel is simpler than ABCD's in that it does not include the offset value $\boldsymbol{c}$, meaning that any covariance which is truly linear but NOT through the origin would require the sum kernel of $LIN + C$. This would be reasonable from a purity-of-model point of view, making the presence of a vertical shift immediately visible in symbolic form (vs seeing $c \neq 0$), but in practice it is needlessly cumbersome computationally and combinatorially when it comes to polynomial kernels, e.g. the kernel space exploration depth of, say, $LIN \times LIN \times LIN$ is shallower than $(LIN + C) \times (LIN + C) \times (LIN + C)$ (though an ad-hoc production rule could shorten it at the price of making rounds more expensive). Secondly, not including the intrinsic $c$ parameter removes the side-effect of being able to simply read out polynomial roots in products of LINs, which is a useful feature on its own, especially when describing fit kernels with text. GPy's PER kernel is in fact the same one from which ABCD's was derived (i.e. MacKay's standard periodic kernel (Duv, 2014)), which suffers from the opposite problem of the above LIN in that it does allow vertical shifts while modelling periodicity, which, given the considerably higher complexity than a simple line, makes it more competitive than it needs to be in fitting data which is not periodic (but has, say, two similar peaks). Specifically, ABCD's PER is the purely-periodic component of MacKay's; given the direct availability of the latter, both were tested and the utility of the former was re-verified: ABCD's PER on its own is able to model, say, $cos(x)$, but not, say, $cos(x) + 1$ (instead requiring the addition of C). While GPy-ABCD uses the ABCD versions of the above two kernels, it does not share the same sigmoidal ones. The original ABCD's SI and

SIr formulae were the same as the S and Sr ones but with the product of opposite-slope sigmoidals in place of every sigmoidal, i.e. $SI(\boldsymbol{x}, \boldsymbol{x'}) = (\sigma(\boldsymbol{x})(1 - \sigma(\boldsymbol{x})))(\sigma(\boldsymbol{x'})(1 - \sigma(\boldsymbol{x'})))$ and $SIr(\boldsymbol{x}, \boldsymbol{x'}) = (1 - \sigma(\boldsymbol{x})(1 - \sigma(\boldsymbol{x})))(1 - \sigma(\boldsymbol{x'})(1 - \sigma(\boldsymbol{x'})))$; there are a few intertwined merits to the versions GPy-ABCD uses:

- the height/depth is fixed to 1 by the denominator and therefore it does not indirectly depend on $(\boldsymbol{x} - \boldsymbol{x'})$ (which would require further scaling by fitting parameters)
- computing the gradients is less computationally intensive
- the window start and end locations are more distinctly identifiable

Being the most complex in the grammar, the CW kernel is unsurprisingly the one requiring the most care in implementation in order to reduce numerical instability and increase result consistency; to this end, many variations of the SI kernel were tried, in both mathematical nature of *sig* and specific parametrisations. In the first respect the tried *sig*s were, in order of decreasing computational complexity, $tanh(x)$, $\frac{x}{\sqrt{1+x^2}}$ and $\frac{x}{1+|x|}$, where the last (and ultimately selected) one has the peculiarity of a very steep derivative culminating in a sharp (removable) discontinuity. In the second respect, some tried parametrisations were start and end locations, central location and width, and start location and width, with the last being ultimately selected; to the same end, in the current version the slope parameter is fixed to a constant, thus reducing the kernel complexity and making it more competitive against those with fewer parameters. An issue has been opened on GPy's repository regarding possibilities of alternated fitting and relative constraining of specific parameters (the two implicit window locations), which would aid convergence in this kernel's fitting machinery. Implementation wise, GPy's `BasisFuncKernel` was used as a base for all sigmoidal kernels, allowing the definition of kernel and gradients through their components, i.e. by $\sigma$, $\overline{\sigma}$, *hat* and *well*, then letting GPy combine them on its own.

### B.2 Kernel Expressions & Simplification

One definition is required before listing the simplification rules:

**Definition 2** *A kernel function is **stationary** if it has no dependence on $\boldsymbol{x}$ and $\boldsymbol{x'}$ except through $\boldsymbol{x} - \boldsymbol{x'}$, meaning that it is not affected by equal shifts in both points*

All base kernels in use here are therefore stationary except for LIN and the sigmoidal ones. The simplification rules are the following (addition and multiplication commutativity, associativity and distributivity apply):

- WN is addition-idempotent, multiplication-idempotent and also acts as multiplicative-zero for stationary kernels, therefore there can be at most one per sum and one with no other stationary factors per product
- C is also addition-idempotent and multiplication-idempotent, but it acts as multiplicative-one, therefore there can be at most one per sum and it is in products only when alone
- SE is multiplication-idempotent
- Since all base kernels include a variance parameter, when in a product they can all be removed in favour of a single product-wide variance

Code-wise GPy-ABCD's design is very different from the original ABCD, having had the benefit of hindsight to globally generalise ad-hoc structures and procedures within a broadly similar algorithm, but there is an important difference in functionality when it comes to expression simplification. The original ABCD has two alternative operating modes which affect expression handling, ABCD-Interpretability and ABCD-Accuracy, the key difference being that the former only works with sum-of-products-form kernel expressions (i.e. it immediately simplifies any nested form, even during model space exploration), while the latter completely avoids this type of simplification, limiting output to nested expressions, which ABCD is not able to rearrange and describe with text. Each mode is named after its advantage, the idea being that forcing exploration to go through specific expression forms does achieve eventual model explainability, but at the price of some bias on the process. GPy-ABCD avoids this bias by possessing the framework to rearrange the outputted symbolic-and-numerical nested expressions, thus retaining both Accuracy and Interpretability: it does not enforce canonical form during model space search (still simplifying as needed) and it only reduces to sum-of-product form at the very end to recover explainability.

## B.3 The Model-Space Search

The main model-space search function exported by the package is:

```
explore_model_space(X, Y,
    start_kernels, p_rules, utility_function, rounds, beam,
    restarts, model_list_fitter, optimiser, verbose)
```

where only `X` and `Y` are required since the rest have default values.

- Every GP regression of `Y` by `X` performs a set number (`restarts`) of random initial-parameter-values restarts in order to increase confidence in having converged to global minima/maxima, and its model score is given by the provided `utility_function`
- The search begins by fitting the provided `start_kernels`, acting as a $0^{\text{th}}$ round, then a set number (`rounds`) of standard rounds is executed, iteratively generating and fitting candidate kernels
- Each round a specific number (`beam`) of the best-scoring kernels from any past round is selected for further processing (previously selected ones are excluded)
- If the `beam` argument is instead a list of integers, then each round will use the corresponding list entry as the beam-search beam. The intent behind this extra configurability is to allow beam narrowing, thus processing more early-round kernels (when expressions are simpler) and fewer late-round ones. When the data shape is particularly complex this avoids premature focus on particular expressions, giving a broader spectrum of simple models a chance, while in simpler cases it instead reduces overall computation time since fewer needlessly complex models are fit
- Every input-kernel is "expanded", meaning that the provided production rules (`p_rules`, see Section B.3) are applied to it, generating new expressions which are then filtered for new and past duplicates before fitting
- Through the `model_list_fitter` and `optimiser` arguments the user has the options of customising, respectively, how to parallelise the fitting of a list of kernels, and which of the fit optimisers available in GPy to use

- If `verbose` is true, intermediate results are printed during execution; currently the search cannot be interrupted and has to complete the predetermined number of rounds
- The returned values are:
  - `sorted_models`: all fit models, ordered by decreasing fit score
  - `tested_models`: a list of lists of fit models, one per round
  - `tested_k_exprs`: the list of all fit kernel expressions
  - `expanded`: the list of all fit models which have been expanded in a round
  - `not_expanded`: the complement of `expanded` with respect to `sorted_models`

GPy-ABCD provides some ready-made lists of starting kernels and production rules, but, more importantly, it also provides the tools to write custom ones; the default ones are:

**Start kernels** all base kernels except for SE (since as a $1^{st}$ round seed it is much too adaptable, obscuring more specific initial ones) plus $2^{nd}$ and $3^{rd}$ order polynomials, a vertically shifted PER and both change-type kernels with LIN as an argument (since they are sufficiently simple cases of each for the purpose of capturing the change pattern early if present). That is: $WN$, $C$, $LIN$, $PER$, $LIN \times LIN$, $LIN \times LIN \times LIN$, $PER + C$, $CP(LIN, LIN)$, and $CW(LIN, LIN)$

**Production rules** using $E$ to indicate any kernel expression and $B$ to indicate a base kernel: the minimal rules to span the base model space ($E \to E+B$, $E \to E \times B$, $B \to B$), simple-case change operators to span the whole model space ($E \to CP(E, LIN)$, $E \to CP(LIN, E)$, $E \to CW(E, LIN)$ and $E \to CW(LIN, E)$), simple expression reductions ($E+E2 \to E$ and $E \times E2 \to E$), and a few heuristic shortcuts to commonly reached forms ($E \to E \times (B + C)$, $E \to B$)

The original ABCD is different in starting kernels, production rules and overall algorithm:

- It has no $0^{th}$ round, and its $1^{st}$ round is seeded by the result of applying the production rules to the simple WN kernel
- The production rules used are the same as GPy-ABCD's except for the absence of the ones discouraging SE in favour of higher-order curves and replacing the change-kernel ones with pairs of both simpler and more complex versions: $E \to CP(E, C)$, $E \to CW(E, C)$, $E \to CP(C, E)$, $E \to CW(C, E)$, $E \to CP(E, E)$ and $E \to CW(E, E)$
- The overall algorithm is not a beam search but a simple greedy search using exclusively the single best model from the previous round to seed the next one, therefore possibly excluding a better model from, say, two rounds before

### B.3.1 Model Selection Method Issue

The choice of utility function with which to score fit models is not free from complications; comparing implementations, the original ABCD uses the Bayesian Information Criterion (BIC), while GPy-ABCD allows the user to provide an arbitrary function and contains a few basic ones (BIC, Akaike Information Criterion (AIC), Akaike Information Criterion corrected for small sample size (AICc) and a Laplace Approximation of Leave-One-Out Cross-Validation error (LA-LOO) (Vehtari et al., 2016)) but currently also defaults to BIC. Putting aside LA-LOO, which does not meet the requirement of using model complexity to balance closeness of fit, there are however some statistical issues with using the above

information criteria, the most important of which being that they all assume the parametric distribution family under consideration contains the true model (Konishi and Kitagawa, 2007), the opposite of which is precisely the usefulness of an ABCD-like model search. A second issue is that these criteria assume the data is independently drawn from said true distribution, while GPs are underpinned precisely by the assumption of ordered correlation. In practice these criteria do a reasonable job of ranking the models, but more statistically sound alternatives (the implementation of which is not straightforward) are worth exploring: Bayesian Predictive Information Criterion (BPIC) (Ando, 2007) and Generalised Information Criterion (GIC) (Konishi and Kitagawa, 2007) explicitly addresses the "true distribution" issue, but other criteria worth examining are Widely Applicable Information Criterion (WAIC) or Widely (Applicable) Bayesian Information Criterion (WBIC) (Watanabe, 2013), and perhaps combinations of complexity penalties with Leave-One-Out error approximations (more likely by empirical justification rather than statistical argument).

## B.4 Model Description

GPy-ABCD's model description procedures cover only the simplest part of the original ABCD implementation's, as the latter produces pages of graphs and parameter analyses for each product in the sum-of-products form. This extended type of output does not match GPy-ABCD's purpose, but its post-search nature means it could be developed as a separate module taking in a kernel expression and its GPy counterpart.

## B.5 Differences Summary

Most differences stem from the core fact that GPy-ABCD is meant to be an open back-end service rather than a full system unto itself; because of this it is modular and developer-friendly, allowing the configurability and extensibility required to perform model searches of arbitrary complexity and constraints. With respect to the scope of analysis and output type, the original ABCD produces documents of tens of pages which include multiple plots, tables and details on how each identified component affects the full model, while GPy-ABCD is only intended to produce developer-friendly model objects and short paragraphs describing them. As for the method, unlike the original ABCD's, GPy-ABCD's kernel space search algorithm allows configurable starting conditions and the expansion-candidates selection is able to indirectly backtrack to previous rounds' results; this means that the explored kernel space can be different from the original system's in starting conditions, evolution steps and weights on the directions of expansion.

# Appendix C. Evaluation Details

The following sections match the evaluation hypotheses in Section 5.

## C.1 Synthetic Data

The evaluation procedure with synthetic datasets is straightforward: produce noisy data from functions exhibiting features the system is meant to be able to capture and verify that the top search results contain the base-kernel version of the used formula. For example: the "obvious" kernel for noisy data produced from $y = 2x \cos(\frac{x-5}{2})^2$ should identify the inter-

action of linearity and of vertically-shifted periodicity, i.e. it should be $LIN \times (PER + C)$. Tests were performed for various combinations of up to 3 base kernels both with and without permanent or temporary kernel changes, and the system successfully and consistently identified each combination with one source of difficulty and one exception:

- $PER$ is able to achieve pathologically competitive scores when fitting complex non-periodic data by converging to periods considerably shorter than any significant data curvature, producing fits which are obviously wrong to a human (a behaviour shared by GPy's own non-purely-periodic kernel version, i.e. MacKay's)
- CW kernels with non-stationary outer parts (e.g. $CW(LIN, \cdots)$; see Section B.2) are almost never identified; this makes sense since the vertical shift induced by the window portion is highly unlikely to match the one required by the non-stationary outer kernel (e.g. a straight line having to stop and then start again exactly at the required height after the window)

Usability-wise the former point is less of an issue for a human user (who can choose any of the top-scoring models based on their own criteria) than it is for a broader framework automating the use of GPy-ABCD; in this case reasonable options are adding a simple closeness-of-fit score filter on the top-scoring models, or one which checks for explicitly periodic features to compare to the identified periods (e.g. a Fourier transform). Addressing the second point is not straightforward since this behaviour is technically correct. Since two nested CP kernels can fit these scenarios at the cost of more parameters, two possibilities come to mind: either this combination could be encouraged by additional production rules, or a new version of CW with an additional vertical shift could be implemented (i.e. a C added to the post-window side of the non-stationary kernel); in both cases the identified parameters of said kernel would however not be reliable (e.g. the roots of a polynomial which is vertically shifted on the right side of the window).

## C.2 Original ABCD Data

Regarding the 2[nd] evaluation point, "similar" system behaviour necessarily has to be reduced to similarity in final fit KEs given the large design differences (see Section B.5). Some datasets with corresponding original-ABCD analyses are publicly available, and, taking the above into account, they can be used to evaluate GPy-ABCD's output in the above respect, but unfortunately not in explored kernel space details or in computational efficiency of single fits (not necessarily useful in any case given the different technology stack). Because the design differences are a core part of what is being evaluated, although it would be possible to make GPy-ABCD run an algorithm which is very close to the original (identical except for not being able to ignore models from previous rounds if better than those strictly in the last one), GPy-ABCD was run on default search parameters with only minor tweaks to compensate for the systems' different intended use cases: GPy-ABCD default settings are for an initially-broad and then narrowing few-round search on reasonably small datasets (100 points or so) whose complexity is not exceptional (the typical data from FRANK's queries is small in size and "shallow" in terms of expression "depth"). For each of the available analyses, the original ABCD's (single-expansion, naive-greedy) model search was run for 10 rounds and with 10 random parameter restarts for each fit model (Lloyd, James Robert et al., 2014). Based on this, GPy-ABCD's search parameters were set to the same

number of restarts and an equivalent number of expansions (though not necessarily of maximum "depth"): 5 rounds with a (non-dynamic) beam of 2. Given the known dataset complexity and to allow fully comparable expressions between systems, the final tweak from GPy-ABCD's default parameters was to re-enable the SE kernel (normally excluded since it tends to dominate early rounds while being the least descriptive). Comparing results, no noticeable differences in closeness of fit were present, and GPy-ABCD's expressions were generally simpler than the original's. This is to be expected since 5 2-beam beam-search rounds are unlikely to be as deep as 10 naive-greedy ones, but a point of note is the frequent use of change-kernels in the original ABCD's, while GPy-ABCD seems to avoid them. At this point it is not clear why this should be the case: assuming equally effective change-kernel fits in both systems (the synthetic data evaluation does not suggest inadequacies on this front), and given the equal number of added parameters (change locations), the complexity penalty in model score (BIC for both) should have the same effect in steering the model-space search focus. It is not inconceivable that the base sigmoidal function difference ($\frac{x}{1+|x|}$ for GPy-ABCD and $tanh$ for the original) could play a part in this, though the precise mechanics are not obvious; differences in fit effectiveness between the two underlying frameworks (Python & GPy vs Matlab & GPML) are an unquantified factor.

### C.3 Effectiveness in Original Purpose

Feature-wise GPy-ABCD is capable of fulfilling everything it was intended for within FRANK, i.e. handling inputs and outputs of requests to fit data of extremely varied shapes, including producing text descriptions, which allows the addition of functional-form-based queries to FRANK's arsenal. However, as the paper so far will have made clear, this type of modelling is not without its downsides, particularly in the context of QA, the principal issue being the high computational expensiveness of the full model-space search (since even a single GP regression with the required multiple random parameter restarts takes some time). GPy-ABCD's configurability-oriented design is in part due to this issue, allowing FRANK's statistics expert system SMART to choose to run it in full only for small datasets (with number of points in the low-hundreds), beyond which it can be run on increasingly more constraining parametrisation profiles (e.g. fewer rounds, production rules or base kernels); alternatively, FRANK could just feed it smaller uniform samples of the datasets to return approximate solutions quickly, and then leave the user the choice of whether to proceed to more expensive models. This level of consideration and weighing of alternatives in the set of available modelling methods (which can overlap over different query types) would be resolved precisely by considerations such as the above, always taking care to inform the user of the reasons and possibly applied restrictions (but also providing the option of force-selecting a path a priori); that is to say that FRANK having to work around this computational expensiveness by offering the user more choices is an acceptable outcome, and one which plays well with other competing modelling components.

