# OpenReview forum: "GPy-ABCD: A Configurable Automatic Bayesian Covariance Discovery Implementation"
_ICML.cc/2021/Workshop/AutoML — AutoML@ICML2021 Poster_

### Official Review · Reviewer_9MXo · 2021-06-07

**Rating:** 7
**Confidence:** 4

**Review:**

The paper describes a software package for Automatic Bayesian Covariance Discovery. The subject is a good fit for the topic of the workshop, and I think will be of interest to other participants.

I have two main comments on how I think the paper could be improved for the future.

1. The motivating use case for the package is the FRANK QA system. I think it'd be nice to provide a bit more context on that application, to provide stronger motivation for the package and a more concrete application. For instance, moving a brief description of FRANK into the Intro might be appropriate.

The most helpful thing would be to see an example of GPy-ABCD applied directly to a FRANK QA problem. The Background gives as a FRANK QA example "How does rainfall in the UK behave over time?". Could you show how GPy-ABCD works on that particular question? What the input data are and what the output description is? I think a concrete application example like that would make the paper more interesting to read and provide stronger motivation.

2. One of the claims in the conclusion is that GPy-ABCD features an "improved model-search algorithm" relative to the original ABCD. There isn't much discussion on this point in the paper. The appendix describes the difference between GPy-ABCD model-search and the original; the main improvement seems to be the beam search vs. the greedy search? I can see how that could be an improvement, but there isn't very strong direct evidence given to support this particular claim. Section 4 describes "no differences in closeness of fit" but "generally simpler" expressions, but I think you'd want to put some more quantitative evaluation of this in order to claim an improvement over the original.

Besides that I just had one minor comment, which was that the description of ABCD as "an unsupervised learning system acting as a framework" doesn't make much sense to me. I'm not sure what it means for a learning system to act as a framework. So that could perhaps benefit from being reworded.

---

### Official Review · Reviewer_kBTR · 2021-06-10

**Rating:** 5
**Confidence:** 4

**Review:**

This paper suggests software, named GPy-ABCD, for solving automatic Bayesian covariance discovery (ABCD). ABCD defines an open-ended language of GP models and produces natural language descriptions of patterns in the data by automatically translating components of the model. This work utilizes the GPy package, which is one of the most popular GP packages.

I would like to describe a few concerns on this work.

1. It does not provide any experimental results in the main article. It is to introduce the ABCD implementation using GPy, but I would like to see whether it works well.
2. It does not mention about competitors in this area. It should be added.
3. This work cites the GitHub repository of GPy-ABCD, but it is strange to me. If this work presents the GPy-ABCD implementation itself, the repository should not be cited.
4. Section numbering is inconsistent. Please add the numbering to Introduction and Conclusion, or eliminate the numbering for all the sections.
5. Writing should be improved. This paper is hard to read.

For the aforementioned reasons, I would like to recommend weak reject for this paper.

---

### Official Review · Reviewer_GMMN · 2021-06-15
**Review: GPy-ABCD: A Configurable Automatic Bayesian Covariance Discovery Implementation**

**Rating:** 7
**Confidence:** 4

**Review:**

The authors present an implementation of an automated Gaussian process modeling strategy using the open source GPy library.  The authors present a reasonable explanation of the algorithm and its goals.  I think that this content is absolutely appropriate for the AutoML workshop.

Two thoughts for the authors

As I read this, I was somewhat confused as to the target audience of this particular library.  I feel like people who want interpretability of their models will usually go with some sort of decision tree.  Most of the time, when I use GP models, I use them in moderate dimensional spaces, e.g., 3-30 dimensions.  I have a mildly hard time envisioning a description of the form presented in Section 3.4 for even more than 1 dimension.  Can that happen?  I would think that being able to do that would be extremely difficult (and, by itself, a rather impressive research accomplishment).

One sentence struck me as odd:
>> Though present and usable, the SE kernel is disabled by default in GPy-ABCD since its
  versatility and small number of parameters make it too competitive against more complex
  but more interpretable expressions.
This feels a bit bizarre.  The mathematical relationship between complexity and interpretability was never explained.  Maybe it was  explained in the original "Automatic Statistician" paper, but this seems to be implying that no possible tradeoff between complexity and interpretability would ever choose another kernel component over the SE kernel.  Is that true?  I would assume that there should be some continuous tunable parameter, somewhere in the decision-making process, which would balance these values appropriately.  How, for example, are the other components balanced, e.g., the very simple Constant kernel versus the presumably more complicated vague "Generalized Periodic" kernel.

---

### Decision · Program_Chairs · 2021-06-21

Accept (Poster)